# Structural Features and Water Resistance of Glass–Matrix Composites in a System of RNO_3_-KHSO_4_-P_2_O_5_ Containing Different Additives

**DOI:** 10.3390/mi14040851

**Published:** 2023-04-14

**Authors:** Alexander Gorokhovsky, Igor Burmistrov, Denis Kuznetsov, Alexander Gusev, Bekzod Khaydarov, Nikolay Kiselev, Elena Boychenko, Evgeny Kolesnikov, Ksenia Prokopovich

**Affiliations:** 1Department of Functional Nanosystems and High Temperature Materials, National University of Science and Technology «MISIS», 119049 Moscow, Russiadk@misis.ru (D.K.); nanosecurity@mail.ru (A.G.); bekzod1991@mail.ru (B.K.); nikokisely12345@gmail.com (N.K.);; 2Department of Materials Chemistry and Chemical Technology, Yuri Gagarin State Technical University of Saratov, 410054 Saratov, Russia; 3Engineering Center, Plekhanov Russian University of Economics, 36 Stremyanny Lane, 117997 Moscow, Russia

**Keywords:** nitrate-sulfate-phosphate glasses, vitrification, structure, water resistance

## Abstract

Low-temperature (350 °C) vitrification in a KNO_3_-NaNO_3_-KHSO_4_-NH_4_H_2_PO_4_ system, containing various additives to improve the chemical durability of the obtained material, was investigated. It was shown that a glass-forming system with 4.2–8.4 wt.% Al nitrate admixtures could form stable and transparent glasses, whereas the addition of H_3_BO_3_ produced a glass–matrix composite containing BPO_4_ crystalline inclusions. Mg nitrate admixtures inhibited the vitrification process and only allowed obtaining glass–matrix composites with combinations with Al nitrate and boric acid. Using ICP and low-energy EDS point analyses, it was recognized that all the obtained materials contained nitrate ions in their structure. Various combinations of the abovementioned additives favored liquid phase immiscibility and crystallization of BPO_4_, KMgH(PO_3_)_3_, with some unidentified crystalline phases in the melt. The mechanism of the vitrification processes taking place in the investigated systems, as well as the water resistance of the obtained materials, was analyzed. It was shown that the glass–matrix composites based on the (K,Na)NO_3_-KHSO_4_-P_2_O_5_ glass-forming system, containing Al and Mg nitrates and B_2_O_3_ additives, had increased water resistance, in comparison with the parent glass composition, and could be used as controlled-release fertilizers containing the main useful nutrients (K, P, N, Na, S, B, and Mg).

## 1. Introduction

The use of nitrogen-containing fertilizers is a key factor in modern agricultural production. However, more than half of N-containing fertilizer is lost to the environment, with impacts on air, water, and soil quality, and biodiversity. Nitrogen loss to the environment contributes to greenhouse gas emissions and climate change. The introduction of a traditional nitrogenous fertilizer, such as potassium nitrate or urea, into an oxide matrix would significantly increase the efficiency of fertilizers, reduce the amount applied to the soil, and as a result decrease air and groundwater pollution [1,2,3].

The slow (controlled)-release fertilizers (CRFs) obtained in this way represent materials that release a proportion of their nutrients not exceeding 100% of the total in the first month of use at a temperature of 25 °C [2,4].

Among SRFs, several vitreous materials have been successfully investigated as environmentally friendly fertilizers. However, most of them represent nitrogen-free phosphate and silica-phosphate glasses based on systems containing SiO_2_, P_2_O_5_, MgO, CaO, K_2_O, Na_2_O, and Fe_2_O_3_ as the main components, which coexist together in an oxide glass structure (i.e., [5,6,7,8,9,10]). Furthermore, alkali–silica–phosphate and alkali–phosphate glasses are made by melting raw material mixtures containing various phosphates, as well as metal nitrates, at temperatures exceeding 500 °C, in order to ensure the decomposition of nitrate and ammonium ions present in the raw materials (i.e., (NH_4_)_2_HPO_4_). However, this approach is not economically acceptable, since it promotes the production of NO_x_ and NH_3_, associated with atmospheric pollution by nitrogen compounds.

Meanwhile, in our previous studies [10], it was shown that meta- and pyrophosphate glasses containing NO_3_^-^ ions in their structure could be produced at low temperatures. It was found that alkali–nitrate–sulfate–phosphate glasses, based on a system of NaNO_3_-KNO_3_-KHSO_4_-P_2_O_5_ and obtained by melting at 350–450 °C, are promising alternative fertilizers, containing important nutrients (N, K, P, S, Na) [11]. However, these previously obtained glasses were characterized with high hygroscopicity, as well as a relatively fast solubility in aqueous solutions, which is inconvenient for application as a controlled-release fertilizer. From this point of view, it is important to look for admixtures that would improve the structural stability of the abovementioned nitrate–sulfate–phosphate glasses. Therefore, the influence of certain additives on the vitrification taking place in a system of NaNO_3_-KNO_3_-KHSO_4_-P_2_O_5_, as well as on the structural features of the obtained products, was investigated.

## 2. Materials and Methods

The batch composition selected to produce the parent nitrate–sulfate–phosphate glass (Table 1) was selected due to its relatively high chemical resistance, in comparison with the vitreous materials investigated in our previous work [11].

The batches containing NH_4_H_2_PO4, KHSO_4_, KNO_3_, and NaNO_3_ with the admixtures selected to improve the durability of the investigated materials (all A.C.S 99+%) were homogeneously ground, progressively heated to 350 °C at a rate of 10 K/min, and maintained at this temperature for 1 h in an alumina crucible. The admixtures of pure H_3_BO_3_, Al(NO_3_)_3_·6H_2_O and Mg(NO_3_)_2_∙6H_2_O, as well as their mixtures, were used to improve the water resistance of the glasses obtained in the selected low-temperature melting glass-forming system.

The additives and the parameters of fusion were selected by taking into account the following factors: Systems based on NH_4_H_2_PO_4_ vitrify at temperatures higher than 200 °C [12]. Al(NO_3_)_3_∙6H_2_O, Mg(NO_3_)_2_·6H_2_O, and H_3_BO_3_ decompose at T > 300 °C, participate in the vitrification process, and can improve the chemical durability of the obtained glasses [13,14,15,16,17].

The batch compositions used to produce the glasses are reported in Table 1. The two kinds of raw material mixture (No 1 and 2) were prepared using the batch of the parent glass composition, with the selected additives applied in different quantities and proportions. Here and subsequently these compositions are marked as 1A and 2A (Al nitrate additive), 1B and 2B (boric acid additive), 1M and 2M (Mg nitrate additive), 1AB and 2AB (Al nitrate and boric acid additives with weight ratio of 1:1), 1AM and 2AM (Al and Mg nitrate additives with a weight ratio of 1:1), 1BM and 2BM (boric acid and Mg nitrate additives with a weight ratio 1:1), 1AB and 2AB (Al nitrate and boric acid additives with a weight ratio of 1:1), and 1ABM and 2ABM (with Al nitrate, Mg nitrate, and boric acid additives with a weight ratio of 1:1:1).

Taking into account certain inaccuracies in the quantification of light chemical elements using EDS analysis, we measured the contents of chemical elements with two methods: EDS and ICP, to control the general chemical composition of the obtained glasses and estimate the systematic error in the determination of [N] in the different vitreous phases. To minimize errors, the EDS analysis was applied with a law acceleration voltage of 5 kV. Furthermore, the glasses produced with the same raw materials at 800 °C for 2 h were also analyzed, to ensure the total absence of nitrogen in their structure and estimate the margin of error of this technique. The abovementioned methodology confirmed that the real contents of nitrogen in the investigated glasses were 1.8 ± 0.2 times higher than the data obtained using EDS analysis at 5 kV.

A scanning electron microscope (SEM) Philips XL30ESEM equipped with EDS analyzer (EDAX Pegasus) was used for energy dispersive X-ray analysis, and the data obtained through plasma-coupling atomic absorption spectroscopy (ICP IRIS Intrepid II XPS, Thermo Scientific, Waltham, MA, USA) was applied to corroborate the results of the X-ray fluorescence spectroscopy.

The thermal stability of the glasses was investigated through DTA (Perkin Elmer/Seiko Instruments, Nishiku, Japan), using a heating rate of 5 °C/min, whereas FT-IR spectroscopy (Nicolet Avantar 320ESP, Madison, WI, USA) was applied to characterize their structure. Whereby, 3 mg of each powdered glass was mixed with 300 mg of KBr and then pressed into pellets of 15 mm diameter, which were used to obtain IR spectra.

The kinetics of dissolution in distilled water, as a function of the chemical composition, was investigated for the produced glasses, in comparison with some traditional nitrogen-containing fertilizers (urea, KNO_3_, and NH_4_H_2_PO_4_). The resulting cylindrical specimens (diameter and height of about 0.8 cm) obtained by quenching the melts (glasses and KNO_3_) into a mold or by compressing the powdered samples (KNO_3_, NH_4_H_2_PO_4_) were used for testing.

To study the release behavior of the obtained vitreous materials, a leaching test was carried out for periods of 1, 5, 10, 15, 20, 25, and 30 days, in accordance with the procedure described in [18]. In this test, cylindrical specimens of the obtained glasses and traditional fertilizers were inserted into vials containing 5 mL of deionized water. Every piece of glass was studied for a certain time period. At the end of every period, 1 mL of the corresponding solution was used for analysis with an inductively coupling plasma instrument (ICP IRIS Intrepid II XPS).

## 3. Results

### 3.1. Vitrification

The data on the influence of the selected additives on the vitrification of the investigated systems at 350 °C are reported in Table 2. Generally, it is possible to note that the admixtures of magnesium nitrate did not favor the vitrification process. The compositions 1AM and 1BM, characterized by relatively low contents of Mg-nitrate, only produced non-transparent opaque glasses. At the same time, the admixtures of Al-nitrate promoted complete vitrification (compositions 1A and 2A) or produced white-colored glasses (compositions 1AM, 1AB, 1AMB, 2AMB), although increased contents of the other admixtures (compositions 2AM) did not allow obtaining any glassy materials.

The general chemical compositions of the obtained glasses could be represented in the traditional form as a combination of oxides or salts; however, taking into account that the structure of these glasses is not yet sufficiently clear, we prefer to represent the chemical composition of the obtained materials as contents of chemical elements (nutrients) or oxides, determined using the ICP method (Table 3). In any case, the obtained materials could be classified as ionic glasses due to a relatively low content of glass-formers.

Taking into account large errors in the determination of the “light” chemical elements content, the data reported in Table 3 has a semi-quantitative character; however, we can assume that the glass–matrix materials obtained in the investigated system at 350 °C contained a sufficiently high quantity of water, similar to other types of low-temperature fused phosphate glasses produced previously [19]. On the other hand, increased quantities of the additives promoted reduced contents of water and almost did not influence the N.

It is necessary to note that the amount of H_2_O in phosphate glasses is important with respect to their properties, since a variation in the amount of structural water influences the physical and chemical properties of the glass, due to its participation in the vitrification process, forming P-O-H-O-P bonds in the glass net [20].

The relatively high content of water in the composition of the produced alkali–phosphate glasses promoted their wettability and solubility in aqueous solutions, which is important for use as a controlled-release fertilizer.

### 3.2. Structure

The SEM micrograph and XRD patterns of the transparent glasses obtained in the system of Al(NO_3_)-KNO_3_-NaNO_3_-KHSO_4_-H_2_O-P_2_O_5_ are reported in Figure 1. These transparent yellowish vitreous materials had a totally amorphous homogeneous structure, without any inclusions with a size of d > λ/4 (λ—wavelength of radiation in the optical range) [19].

The white color of the glasses obtained with the compositions No 1B, 1AB, 1AMB, 2B, and 2AB indicated the presence of other phases in their structure. In accordance with the XRD and SEM analysis data, it was possible to identify the presence of BPO_4_ crystalline phase in all the glasses produced with the H_3_BO_3_ admixtures, as well as liquid phase immiscibility taking place in the glass-forming melts of the 1AB and 2AB compositions (Figure 2).

The EDS analysis of the different vitreous phases indicated the presence of all the chemical elements in the main glassy matrix as vitreous inclusions present in the structure of samples 1AB and 2AB (Figure 2d). However, in spite of large errors in the EDS analysis of light chemical elements, it was possible to note that the dark vitreous inclusions had higher contents of B and Al (two-times higher) and lower content of N (four-times lower).

The introduction of Mg in the investigated glass-forming systems only favored vitrification with the compositions 1AM, 1AMB, and 2AMB. The obtained product contained crystalline inclusions (Figure 3a) such as KMg_1−x_H_x_(PO_3_)_3_ (x = 0–0.3) [21] and some unidentified crystalline phases. In the structure of the boron containing sample 1AMB, there were some inclusions of BPO_4_ crystals [22], similarly to in samples 1B and 2B (Figure 2a).

It is interesting that all the Mg-containing glasses showed a trend of meta-stable liquid phase immiscibility, which favored the formation of multiple totally vitreous inclusions with different chemical compositions (Figure 3b,c and Table 4). At the same time, the main glassy matrix had an inhomogeneous structure, due to irregularly dispersed small-sized unknown crystalline inclusions.

The low-energy EDS analysis indicated that the main glass–crystalline matrix contained all the chemical elements present in the obtained materials (point 1, Figure 3d, Table 3), whereas the chemical composition of the vitreous inclusions varied in a wide range. Some glassy droplets were enriched with S and Na (point 4, Figure 3d, Table 3), while others were characterized with relatively high contents of B and Mg (point 2, Figure 3d, Table 3) or Al (points 2 and 3, Figure 3d, Table 3). It is important that almost all the glassy phases had sufficiently high contents of N. Taking into account the results of the comparative analysis using ICP and low-energy EDS, which indicated that the nitrogen concentrations determined with EDS (point 1) were reduced by ~1.5-times their real values, it was possible to assume that the main matrix contained ~6 at.% nitrogen, whereas the nitrogen contents in the glassy inclusions of the main matrix were characterized by reduced values of [N] (~3 at.%, points 2, 3, and 4).

The IR spectra of the materials obtained in the RNO_3_-KHSO_4_-P_2_O_5_ system with the Al-, B-, and Mg-containing additives (Figure 4) show that these admixtures did not significantly influence the glass-forming matrix structure of the parent glasses [11]. Pyrophosphate chains and insulating metaphosphate structural units, as well as orthophosphate anions, were identified from the characteristic bands of the FT-IR spectra [12,23].

The obtained FT-IR spectra also presented adsorption bands corresponding to the bending and stretching modes in the SO_4_^2−^ ions (620 and 1150 cm^−1^, respectively) [24], as well as stretching modes of the N=O chemical bonds in the NO_3_^-^ ions (1410 cm^−1^) and weak bending modes in the NH_4_^+^ ions, confirming their presence in the structure [25].

The ^31^P NMR spectra (Figure 5) confirmed the presence of different phosphate configurations in the obtained materials. In accordance with [26], the weak resonance signals located at 0.47 and 0.87 ppm, corresponded to isolated orthophosphate groups surrounded either by H_2_O molecules and/or by NO_3_^−^ and SO_4_^2−^ ions (Q^0^ group). The resonance peaks located at −12.0 and −12.3 ppm corresponded to the chain ends or pyrophosphate structures (Q^1^ group); meanwhile, the signals at +25.9 and +26.0 ppm corresponded to metaphosphates (Q^2^ groups).

The NMR resonance bands could be used to estimate the relative concentrations of the metaphosphate, pyrophosphate, and orthophosphate structural units in the obtained glasses (Table 5). The NMR spectra indicated that the glass–matrix composite based on composition 1AMB, produced with lower contents of additives, was characterized with a larger amount of metaphosphate chains and a lower content of insulated orthophosphate ions.

### 3.3. Chemical Durability

The kinetic data on the weight losses of the different investigated glass–matrix composites are reported in Figure 6.

The obtained results indicated that the produced glass–matrix materials had improved water resistance, in comparison with some traditional nitrogenous fertilizers (KNO_3_ and NH_4_H_2_PO_4_) and the parent glass composition (number 0 in Table 1). The introduction of B_2_O_3_ into the glass–matrix structure (compositions 1B and 2B) only slightly increased the chemical resistance in water, independently of the B_2_O_3_ contents. At the same time, the aluminum-containing materials increased the chemical durability quite significantly.

The simultaneous presence of boron and aluminum in the glass–matrix compositions provided the greatest increase in chemical resistance. It is also important to note that the material 1AMB had a slightly better water resistance in comparison with the composite 2AMB.

## 4. Discussion

The obtained results showed that H_3_BO_3_, as well as Al and Mg nitrates and their mixtures, introduced in the batches of the parent glass composition did not influence the vitrification in the (K,Na)NO_3_-KHSO_4_-NH_4_H_2_PO_4_ system at 350 °C. However, the glassy materials produced had a new structure, depending the nature of the additives.

The introduction of Al nitrate produced homogeneous one-phase glasses (1A and 2A), similar to the parent glass-forming system (Figure 1). In our case, it was possible to assume that the Al nitrate, introduced in the glass composition by melting at 350 °C, broke down at T > 200 °C and generated amorphous alumina [13], which formed a sufficiently stable glass network structure, similar to that observed in the aluminophosphate glasses produced at higher temperatures [27]. In any case, the introduction of Al_2_O_3_ in the structure of the alkali–phosphate glasses led to the formation of water-resistant Al–O–P bonds, which replaced the easily hydrolysable P–O–P links and improved the chemical durability of the modified phosphate glasses.

On the other hand, the [AlO_4/2_] tetrahedra [27] favored the connection of the end groups of pyrophosphates (Q1) and increased the content of methaphosphate (Q^2^) structural units (Figure 4 and Figure 5). As a result, the obtained glassy matrix had a higher degree of crosslinking, was less prone to hydrolysis, and therefore had a higher water resistance (Figure 6).

The ways in which sulfate can be incorporated into the phosphate network via S-O-P bonding were considered previously [28,29]. It was established that the presence of sulfates in ZnO–Na_2_O–SO_3_–P_2_O_5_ and Li_2_SO_4_–LiPO_3_ glass-forming compositions led to an increase of water resistance [30], despite the reduction of the amount of glass formed [31].

In accordance with the ICP analysis data, all the obtained materials contained nitrogen (N) in their glassy matrix. Taking into account the relatively high stability of NO_3_^−^ ions and impossibility of forming an oxynitride glass network under these conditions [32], it is possible to conclude that N was incorporated into the matrix in the form of NO_3_^−^.

The FT-IR spectra confirmed that the SO_4_^2−^ and NO_3_^−^ were incorporated into the structure of the obtained glassy matrices, as anions. Thus, we can assume that their location in the glass network could be similar to the case of typical ionic glasses, such as K_0.67_Ca_0.33_(NO_3_)_1.33_ [19,33].



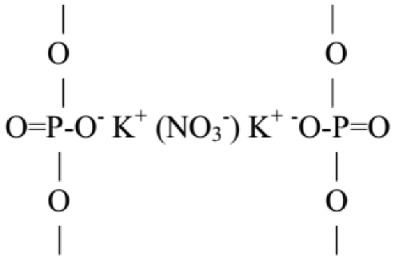



The introduction of B_2_O_3_ into the parent glass composition (1B, 2B) also increased the chemical resistance of the glass–matrix composites, similarly to in the case of alkali–phosphate glasses [14,15]. However, some part of B participated in the formation of the BPO_4_ crystalline phase (Figure 3) and supported liquid phase immiscibility in the glass-forming melts (Figure 2 and Figure 3). The droplets of additional glassy phases that appeared in the main matrix could be enriched or depleted in boron (Figure 3, Table 4). It is known that if the content of SO_4_^2−^ exceeds certain limits, phase separation and crystallization take place in phosphate glasses [34,35]. Similarly, lithium sulfoborate glasses also show a trend of liquid phase immiscibility and crystallization at high sulfate contents [11]. Thus, it is possible to propose that the introduction of B_2_O_3_ in the parent glass compositions containing SO_4_^2−^ supported both the abovementioned processes.

In addition, it is possible to note that the introduction of Mg-nitrate also favored crystallization and liquid phase separation in the parent glass compositions. KMgH(PO_3_)_3_, with some unidentified crystalline phases being recognized in the glassy product based on the 1M and 2M compositions, as well as 1AM, 1AMB, and 2AMB (Figure 3). Furthermore, in the batches containing H_3_BO_3_ and/or Al nitrate, the introduction of Mg nitrate additive, decayed at T > 300 °C [16], inhibited the vitrification processes (Figure 3); the compositions with high contents of Mg nitrate (2AM, 2AMB) did not vitrify as well as the compositions free from Al-nitrate additives (1BM and 2 BM).

In any case, due to the abovementioned reasons, the glass–matrix materials containing all three kinds of investigated additive (1AMB and 2AMB) were characterized by intensive liquid phase separation and multiple inclusions of various crystalline phases (Figure 3). The presence of Al nitrate in the raw material mixtures supported the vitrification process and stabilized the glass-forming network of the parent phosphate glass. The glassy-matrix composites based on the raw material mixtures 1AMB and 2AMB had a sufficiently high water resistance, similar to that of other known controlled-release fertilizers [1,2,3,4,5,6,7,8,9,10]. However, the developed compositions had a greater number and higher content of useful nutrients (P, K, Na, S, N, B, Mg) and could be considered promising nitrogen-containing controlled-release fertilizers (CRF).

## 5. Conclusions

The glass–matrix composites obtained in this research, using raw material mixtures based on the system RNO_3_-KHSO_4_-P_2_O_5_ (R = K, Na) and containing different additives (H_3_BO_3_, Al and Mg nitrates), had a slow rate of dissolution in water, controlled by the quantity and composition of the admixtures. Such materials are of particular importance in modern agricultural technologies, to use them as slow (controlled)-release fertilizers and containing all the main nutrients (K, P, N), as well as some micronutrients (Na, B, Mg). The water resistance of these composites is determined by their specific microstructure. The main phosphate glassy matrix, which has a moderate chemical durability, can become degraded under the action of water, with the progressive release of useful nutrients. This process is accompanied by the introduction of various microscale vitreous and crystalline inclusions in the solution, present in the matrix. These vitreous micro-inclusions also degrade under the action of water; however, their chemical compositions differ. As a result, the dissolution of such glassy droplets in water has a prolonged character. The crystalline micro-inclusions, also containing useful nutrients, have different rates of dissolution in aqueous solutions. Thus, the mechanism of nutrient release in the obtained glass–matrix composites is similar to the drug release in long-acting medicines.

## Figures and Tables

**Figure 1 micromachines-14-00851-f001:**
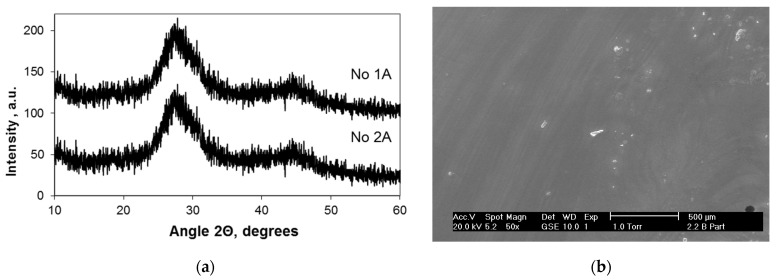
XRD patterns (**a**) and SEM micrograph (**b**) of the materials 1A and 2A (**b**).

**Figure 2 micromachines-14-00851-f002:**
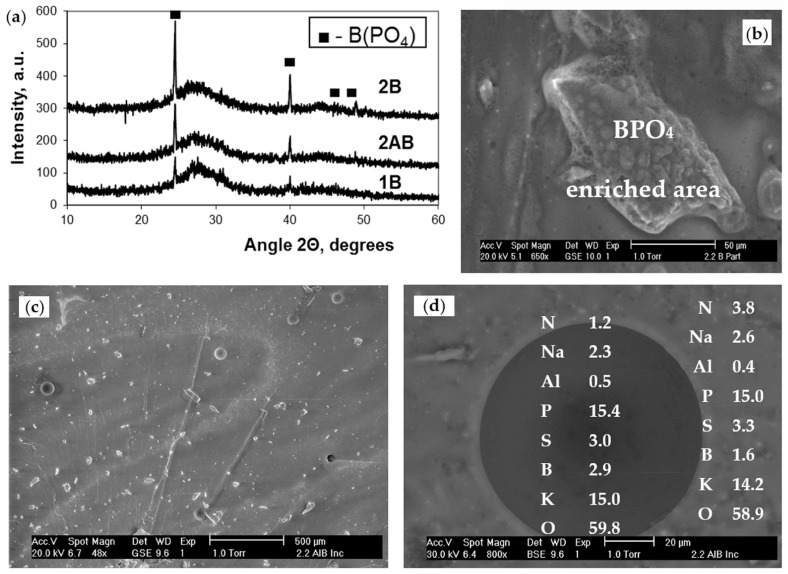
XRD patterns of the glasses 1B, 2B, and 1AB (**a**) and electron images of their structure: 2B (**b**) and 1AB (**c**,**d**). The chemical composition of the vitreous phases (droplet and matrix) is reported in accordance with the data from the low-energy EDS analysis (without hydrogen).

**Figure 3 micromachines-14-00851-f003:**
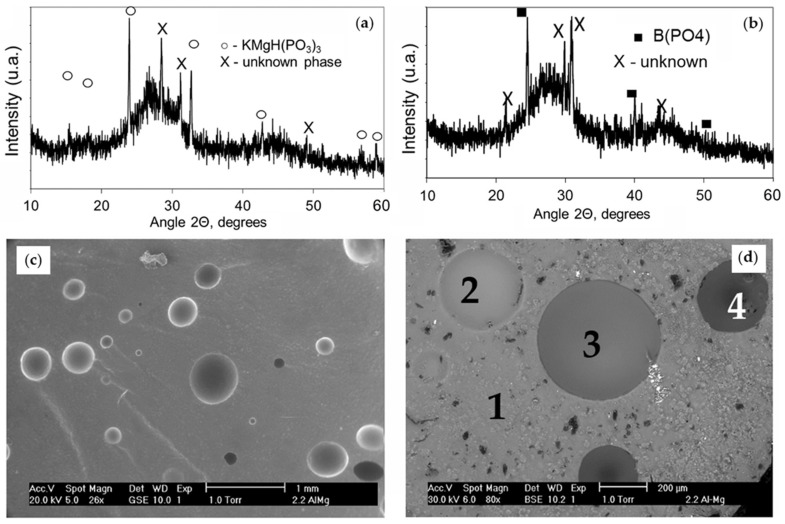
XRD patterns (**a**,**b**) and electron images (**c**,**d**) of the materials based on the compositions 1AM (**a**,**c**) and 1AMB (**b**,**d**). X—unknown crystalline phases. Chemical composition in point 1–4 presented in the Table 4.

**Figure 4 micromachines-14-00851-f004:**
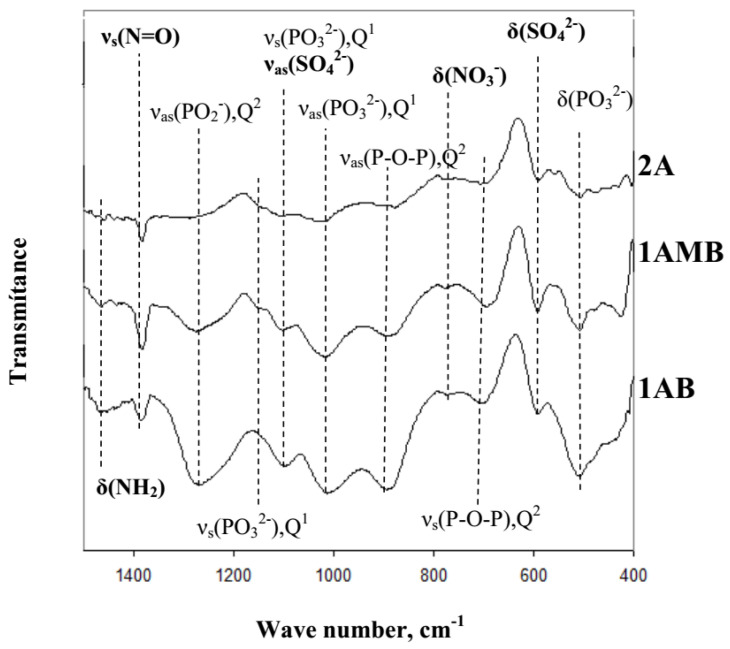
FT-IR spectra of the materials 1A, 2AB, and 1AMB.

**Figure 5 micromachines-14-00851-f005:**
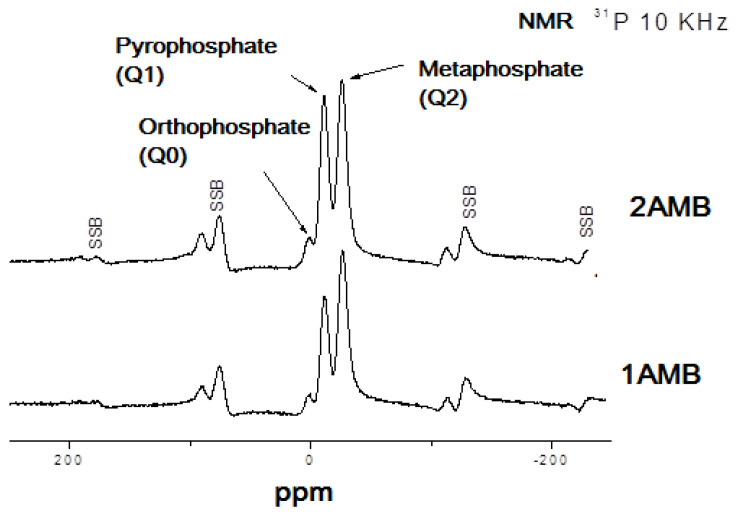
^31^P NMR spectra of the samples 2AMB and 1AMB. SSB—spinning side bands.

**Figure 6 micromachines-14-00851-f006:**
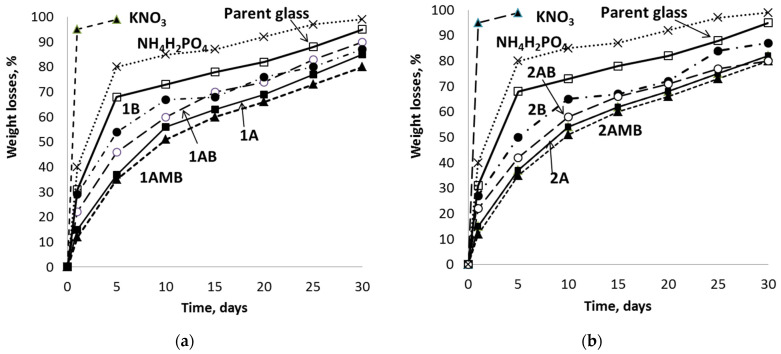
Kinetic curves of the dissolution of the prepared glass–matrix materials in water, in comparison with some traditional nitrogenous fertilizers. Materials based on the batches containing 4.2 wt.% (**a**) and 8.4 wt. % (**b**) of the Al, B, and Mg containing additives. The marks correspond to the numbers in Table 2.

**Table 1 micromachines-14-00851-t001:** Raw material mixtures used in the experiments. The contents of Al and Mg nitrates are indicated, excluding the H_2_O contained in their hydrates. The mixture 0 is a batch composition of the parent glass.

No	Content of the Component (wt.%)				
	NH_4_H_2_PO_4_	KNO_3_	NaNO_3_	KHSO_4_	{H_3_BO_3_ + Al(NO_3_)_3_ + Mg(NO_3_)_2_}
0	67.7	21.3	7.4	4.9	-
1	64.8	20.4	7.1	3.5	4.2
2	62.1	19.4	6.7	3.4	8.4

**Table 2 micromachines-14-00851-t002:** Influence of the different admixtures on the vitrification in the system investigated: Al(NO_3_)_2_∙6H_2_O (A), Mg(NO_3_)_2_∙6H2O (M), and H_3_BO_3_ (B).

Group	Admixtures (Glass Composition Number)						
	Al	Mg	B	Al + Mg	Al + B	Mg + B	Al + Mg + B
1	Transparent glass (1A)	Non (1M)	White glass (1B)	White glass (1AM)	White glass (1AB)	Non (1BM)	White glass (1AMB)
2	Transparent glass (2A)	Non (2M)	White glass (2B)	Non (2AM)	White glass (2AB)	Non (2MB)	White glass (2AMB)

**Table 3 micromachines-14-00851-t003:** Chemical composition of various glass–matrix composites produced (combined data of ICP and EDS analysis).

**No**		**Composition (Chemical Elements, at.%)**								
	**H**	**N**	**S**	**K**	**Na**	**P**	**Al**	**B**	**Mg**	**O**
0	8.2	8.1	4.4	14.0	2.8	14.7	-	-		47.7
1AMB	6.9	7.8	4.9	14.1	3.1	15.5	2.1	1.8	0.2	44.6
2AMB	6.4	7.3	4.6	14.6	2.7	14.6	2.7	1.9	0.4	44.8
	**Composition (oxides, mol.%)**									
	**H_2_O**	**N_2_O_5_**	**SO_3_**	**K_2_O**	**Na_2_O**	**P_2_O_5_**	**Al_2_O_3_**	**B_2_O_3_**	**MgO**	
0	12.2	14.1	15.4	24.3	4.9	29.1	-	-	-	
1AMB	11.5	12.8	15.5	23.0	4.8	25.1	3.6	3.0	0.7	
2AMB	10.5	12.2	15.0	23.6	4.6	25.6	4.3	3.3	0.9	

**Table 4 micromachines-14-00851-t004:** Chemical composition of different vitreous phases in glass 1AMB, in accordance with the data of low-energy EDS analysis (at.%, without oxygen). The points are marked in Figure 3.

Chemical Elements	Relative Contents, at.%			
	Point 1	Point 2	Point 3	Point 4
N	4.1	2	1.6	2.5
B	0.8	1.4	0.3	0.2
Na	2.6	2.8	2.6	3
Al	0.2	7.9	4.6	0.4
P	14.6	11.7	12.7	14.9
S	4.4	4.1	5.4	6
K	14.3	10.4	13	14
Mg	0.2	0.6	0.3	-
O	58.8	59.1	59.5	59

**Table 5 micromachines-14-00851-t005:** Relative contents of different structural phosphate units in the materials, based on the compositions 1A and 1AMB.

Configuration	Content, mol.%	
	2AMB	1AMB
Orthophosphate	9.4	4.5
Pyrophosphate	48.2	41.7
Metaphisphate	42.4	53.8

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
