# Peer review of "Structural Features and Water Resistance of Glass–Matrix Composites in a System of RNO3-KHSO4-P2O5 Containing Different Additives"

_micromachines, 2023, doi:10.3390/mi14040851_

Round 1

Reviewer 1 Report

In this article the authors demonstrate the low temperature (350oC) vitrification in the KNO3-NaNO3-KHSO4-NH4H2PO4system containing some additives to improve the chemical durability. It is also shown that the glass-matrix composites based on the (K,Na)NO3-KHSO4-P2O5 glass forming system containing Al and Mg nitrates and B2O3 additives have increased water resistance, in comparison with the parent glass composition, and could be used as controlled release fertilizer containing the main useful nutrients (K, P, N, Na, S, B, Mg).

The article is interesting and can be informative for the readers. I would like to suggest some points before acceptance.

1. I would like to suggest authors, if they can go for contact angle measurement for the different types of glasses, to know the measure of the wettability of a synthesized glasses by a water.

2. I am curious to know if authors check the transparency of the glasses by optical methods, by using the different additives, how the transparency is varying, it will be interesting.

3. I see a lot of typo errors in the manuscript, please check carefully the manuscript.

Author Response

Dear colleague, thanks a lot for your comments and recommendations! We have tried to take them into account in our corrections.

Unfortunately or fortunately the synthesized alkali-nitrate-sulfate-phosphate glasses are characterized with a very high wettability by water due to a presence of H2O in their structure. This fact promotes as high wettability by aqueous solutions as relatively high rate of their dissolution in water. This paper describes possible improvement of water resistance of the parent glass composition by using different additives to the batch.

This aim was achieved, however, taking into account possible application of these glasses as slow release fertilizers, the obtained (improved) glasses have to soluble in water too. That is why it is very difficult to determine their wettability correctly by a contact angle measurement.  The glass surface begins interact chemically with water immediately after the contact. When, the glasses of different chemical compositions promote quick change of the chemical composition of the contacted aqueous solution which influences a wettability.

Nevertheless, we introduced into the text some comments on this item.

We used a term ‘transparent glass” only taking into account a possibility to read the text through a glassy plate. This property indicates a presence or absence of any structural inclusions  with d> λ/4 in the glass matrix.

The optical measurements are possible but are not interesting if we will consider these materials as potential slow release fertilizers.

However, we have introduced some comments on transparency of the obtained glasses to specify the above mentioned features.

The typo errors were corrected

Reviewer 2 Report

MY REVIEW 

Structural features and water resistance of glass-matrix composites in the system of RNO3-KHSO4-P2O5 containing different additives

Glass composites is a term for a composite material consisting primarily of glass fibres that have been impregnated with a polymer resin. The components connected in this way create a material that is durable, lightweight and corrosion resistant. Due to their advantages, glass composites are used in more and more industries. Their favourable strength to weight ratio allows them to be used in constructions for the transport industry - trains, buses, cars and others.

Implementing the assumptions of the work Structural features and water resistance of glass-matrix composites in the system of RNO3-KHSO4-P2O5 containing different additives, low-temperature (350oC) glass transition in the KNO3-NaNO3-KHSO4-NH4H2PO4 system containing additives improving the chemical durability of the obtained material was tested.

It was found that alkali-nitrate-sulphate-phosphate glasses (based on the NaNO3-KNO3-KHSO4-P2O5 system), obtained by melting at 350-450 ºC, are promising alternative fertilizers containing important nutrients such as N, K, P . S. Na. However, the previously obtained slides were characterized by high hygroscopicity, as well as relatively fast solubility in aqueous solutions, which was inconvenient to use as controlled-release fertilizers.

In the reviewed work, the composition of the set selected for the production of parent nitrate-sulphate-phosphate glass is presented in Table 1. It should be noted that the composition of the set was selected due to its relatively high chemical resistance compared to other glass materials tested earlier.

Data on the effect of selected additives on the glass transition of the tested systems at 350ºC are presented in Table 2. It can be seen that the admixtures of magnesium nitrate were not conducive to the vitrification process (glass transition - kinetic transformation of the transition from the liquid state to the glassy state). Compositions 1AM and 1BM having a low content of Mg nitrate gave only opaque glasses. The admixtures of aluminium nitrate favoured the completion of the glass transition, or the formation of several white glasses (compositions 1AM, 1AB, 1AMB, 2AMB).

SEM micrograph and XRD diffractogram of transparent glasses obtained in the Al(NO3)-KNO3-NaNO3-KHSO4-H2O-P2O5 system are shown in Fig. 1. Transparent glassy materials have an amorphous homogeneous structure (without any inclusions). Based on the data from the XRD and SEM analysis, the presence of the BPO4 crystalline phase in all glasses produced with H3BO3 admixtures and the immiscibility of the liquid phase (occurring in glass melts of compositions 1AB and 2AB, Figure 2) can be confirmed. EDS (Energy Dispersive X-ray Spectroscopy) analysis of various glassy phases indicates the presence of all chemical elements, both in the main glassy matrix and in glassy inclusions (Fig. 2d).

Low-energy EDS analysis indicates that the main glass-crystalline matrix contains all the chemical elements represented in the obtained materials (Fig. 3d, Table 3), while the chemical composition of the glassy inclusions varies over a wide range.

IR spectra of materials obtained in the RNO3-KHSO4-P2O5 system with additives containing Al, B and Mg (Fig. 4) show that these admixtures do not significantly affect the structure of the glass-forming matrix of the parent glasses.

While implementing the assumptions of the work, the authors also studied the chemical durability, kinetic data on mass losses of various tested glass matrix composites (Figure 6).

The obtained results indicate that the obtained glass matrix materials have better water resistance compared to some traditional nitrogen fertilizers (KNO3 and NH4H2PO4) and the parent glass composition. The introduction of B2O3 to the structure of the glass matrix (compositions 1B and 2B) only slightly increases its chemical resistance in water, regardless of the B2O3 content.

Taken together, the results obtained showed that H3BO3 and Al and Mg nitrates as well as their mixtures introduced into the batch of host glass composition did not affect the glass transition. The obtained glassy materials have an original structure depending on the nature of the additives.

In each case, the introduction of Al2O3 into the structure of alkaline phosphate glasses leads to the formation of watertight Al–O–P bonds.

According to the ICP analysis data, all obtained materials contain nitrogen (N) in their glassy matrix. The introduction of B2O3 into the composition of the host glass (1B, 2B) also increases the chemical resistance of the glass-matrix composite, as in the case of alkali-phosphate glasses.

In any case, for the reasons mentioned above, glass matrix materials containing all three types of tested additives (1AMB and 2AMB) are characterized by intensive liquid phase separation and multiple inclusions of different crystalline phases (Fig. 3).

The presence of Al nitrate in raw materials supports the vitrification process and stabilizes the glass-forming lattice of the parent phosphate glass. Glass matrix composites are characterized by a sufficiently high water resistance, similar to other known controlled-release fertilizers.

It should be emphasized that the developed compositions have a higher number and higher content of useful nutrients (P, K, Na, S, N, B, Mg) and can be considered as a "promising" controlled release nitrogen fertilizer (CRF).

In the paper, the cited references (35) are consistent with the research topic being pursued - structural features and water resistance of glass composites (tested in the RNO3-KHSO4-P2O5 system with various additives). The analysis, discussions and interpretations of the test results presented in figures (6) and tables (5) are carefully made.

Attention should also be paid to the rightness of the work carried out, nitrogen is the most important nutrient for plants and is taken up by them in the largest amounts. This element plays a key role in the growth and development of plants.

According to the reviewer, the work on Structural features and water resistance of glass-matrix composites in the system of RNO3-KHSO4-P2O5 containing different additives and the results obtained on the use of nitrogen-containing fertilizers (a topic that is one of the key factors of modern agricultural production) is highly significant. Nitrogen is one of the most yield-forming of all micro- and macro-forming ingredients needed by plants.

These conducted research are of particular importance in modern technologies for the further development of science and economy. This fact should be particularly emphasized in the summary of the work, which is also essential in the implementation of scientific work/research.

The reviewed work also requires careful linguistic correction (shorter sentences, then the work is more readable and understandable). After completing these reservations, the work Structural features and water resistance of glass-matrix composites in the system of RNO3-KHSO4-P2O5 containing different additives is suitable for printing.

Author Response

Dear colleague, thanks a lot for your comments and recommendations! We have tried to take them into account in our corrections. All remarks are taken into account in the text of the manuscript. Edits and additions are highlighted in color. Also, please see the attached file.

Round 2

Reviewer 1 Report

The authors have modified the manuscript according to the suggestions.